# Photosynthesis in the Biomass Model Species *Lemna minor* Displays Plant-Conserved and Species-Specific Features

**DOI:** 10.3390/plants12132442

**Published:** 2023-06-25

**Authors:** Monique Liebers, Elisabeth Hommel, Björn Grübler, Jakob Danehl, Sascha Offermann, Thomas Pfannschmidt

**Affiliations:** Pflanzenphysiologie, Institut für Botanik, Naturwissenschaftliche Fakultät, Gottfried-Wilhelm-Leibniz-Universität Hannover, Herrenhäuser Str. 2, 30419 Hannover, Germany; m.liebers@botanik.uni-hannover.de (M.L.); elisabeth.hommel@web.de (E.H.); bjoern.gruebler@gmx.de (B.G.); j.danehl@botanik.uni-hannover.de (J.D.); s.offermann@botanik.uni-hannover.de (S.O.)

**Keywords:** *Lemna minor*, *Arabidopsis thaliana*, photosynthesis, post-translational modifications, photosystem antenna, photoinhibition

## Abstract

*Lemnaceae* are small freshwater plants with extraordinary high growth rates. We aimed to test whether this correlates with a more efficient photosynthesis, the primary energy source for growth. To this end, we compared photosynthesis properties of the duckweed *Lemna minor* and the terrestrial model plant *Arabidopsis thaliana*. Chlorophyll fluorescence analyses revealed high similarity in principle photosynthesis characteristics; however, *Lemna* exhibited a more effective light energy transfer into photochemistry and more stable photosynthesis parameters especially under high light intensities. Western immunoblot analyses of representative photosynthesis proteins suggested potential post-translational modifications in *Lemna* proteins that are possibly connected to this. Phospho-threonine phosphorylation patterns of thylakoid membrane proteins displayed a few differences between the two species. However, phosphorylation-dependent processes in *Lemna* such as photosystem II antenna association and the recovery from high-light-induced photoinhibition were not different from responses known from terrestrial plants. We thus hypothesize that molecular differences in *Lemna* photosynthesis proteins are associated with yet unidentified mechanisms that improve photosynthesis and growth efficiencies. We also developed a high-magnification video imaging approach for *Lemna* multiplication which is useful to assess the impact of external factors on *Lemna* photosynthesis and growth.

## 1. Introduction

The duckweed *Lemna minor* is a member of the *Lemnaceae*, a small family of cosmopolitan aquatic plants found in freshwater ecosystems consisting of five genera with 36 species in total [1]. Despite their simple two-dimensional morphology comprising a small number of leaves floating on the water surface and a species-specific number of roots, they are true vascular plants belonging to the group of monocotyledons closely related to the *Araceae*. *Lemnaceae* usually multiply by vegetative propagation without sexual recombination, although they are able to flower and to sexually reproduce under specific conditions. They typically grow in calm freshwater ponds or lakes and, under ideal conditions, can exhibit extraordinary growth rates with biomass doubling times of less than 48 h, rendering them the fastest growing plants on Earth [2]. The simple body structure of *Lemnaceae* can be regarded as the end of an evolutionary reduction process that represents an adaptation of formerly terrestrial plants to an aquatic floating life style. At the same time, the simple frond structure provided an ideal base for an evolution towards vegetative propagation by fission. This process eventually led to the preferred multiplication mode of *Lemnaceae* because it turned out to be highly beneficial for geographical distribution and survival of the species [3]. These particular properties of *Lemnaceae* make them highly interesting for in vivo research approaches targeting plant biochemistry or ecophysiology including studies on photosynthesis. Further, they are suitable for a plethora of biotechnology applications covering, for instance, the production of food for cattle and humans, production of biofuels as well as a high variety of biomolecules such as vitamins, carotenoids, starch (and many others), or as bioassay organisms in toxicity tests and also in waste water management [3,4,5].

Because of its high biotechnology potential, the research interest in *Lemnaceae* has largely increased in the last 10 years. Most potential applications are directly or indirectly connected to the extraordinary growth properties of these plants. However, the reasons why *Lemnaceae* exhibit these high growth rates remain largely not understood until now, especially at the molecular level. Potential reasons discussed include their special ecological niche with unlimited water supply, the high availability of nutrients dissolved in the water column or the reduced limitation in light harvesting through competition for light with neighboring plants or through self-shading. The primary energy source for growth in all plants is photosynthesis. The high growth potential of *Lemnaceae* thus could be associated with specific molecular properties, e.g., in photosynthesis, not yet identified. The *Lemnaceae* have been used for many ground-breaking studies in plant biochemistry and physiology in the 1950–1990s including photosynthesis research [6,7,8] but were largely neglected afterwards with the rise of genetically more easily tractable model systems, such as *Arabidopsis thaliana* or *Chlamydomonas reinhardtii* [3]. Many aspects of our understanding of photosynthesis in *Lemnaceae,* thus, are either technically out of date or have never been investigated in depth. In addition, the recent advances in genome sequence information provide potential novel information not available in the early studies on photosynthesis of *Lemnaceae* [9].

A recent study demonstrated that *Lemna gibba* displays a high plasticity in its response to changing growth light intensities, indicating that this duckweed can easily adapt to highly varying light intensities [10]. There are recent reports indicating that *Lemna minor* can also grow effectively over a range of light conditions [11,12]. Here, we have studied a number of important physiological photosynthesis parameters under a wide range of light intensities as well as selected molecular properties of the photosynthetic apparatus of *Lemna minor* by a combination of in vivo and in vitro approaches. We report commonalities as well as specific differences to the terrestrial model organism *Arabidopsis* which may provide first hints on evolutionary molecular adaptations of photosynthesis potentially associated with the high growth rates of *Lemnaceae*.

## 2. Results

**Imaging of *Lemna minor* growth**: we have grown *Lemna minor* under sterile conditions on liquid medium as described [13]. Because sugars are known to induce inhibitory effects on photosynthesis gene expression in plants [14], we strictly avoided the addition of any carbon source in our growth media in order to assess the true photosynthesis potential of *Lemna minor*. To this end, *Lemna minor* fronds were pre-cultured and maintained in the exponential growth phase by regular transfer between bottles of growth media (Figure 1A). Fronds from these cultures were then used for further analyses. For physiological photosynthesis measurements, *Lemna* fronds were transferred to beakers or Petri dishes with identical growth medium but in non-sterile conditions that allowed the removal of the lid for imaging of chlorophyll (Chl) fluorescence from the top. *Lemna* fronds swimming on liquid medium, however, always showed some floating caused by micro-convections in air and medium that occurred even in a growth chamber. This floating often resulted in disturbances of the detected Chl fluorescence signals interfering with a trustful Chl fluorescence analysis. Prevention of floating could be achieved by filling the measurement vessel completely with *Lemna* fronds in such a way that the plants have no further space for any movement (Figure 1B). However, *Lemnaceae* are known to be touch-sensitive and stop growing once a certain density is reached. Because this may have a feedback effect on photosynthesis, we tested other ways to prevent the floating. For short-term imaging (in the range of min to a few hours), we prevented any movement by pinching the roots of single fronds with a layer of staples on the bottom of a Petri dish. For long-term imaging (many hours to days) that may be disturbed in addition by growth-related movements, we found the use of self-made plastid grids very efficient. It allowed growth of the roots and also of new fronds but prevented very effectively any growth-induced lateral movements and was suitable to record qualitative long-term high-magnification videos of *Lemna* fronds during growth (Figure 1C,D and Appendix A). This setup was found to exhibit a very high positional stability of single fronds even allowing the observation of stalk formation between mother and daughter fronds (Figure 1D, white arrows). It furthermore allowed the direct determination of growth rates by detecting the recorded leaf area using ImageJ software (Figure 1E) revealing growth rates comparable to those reported on the base of frond number [2].

**General characteristics of *Lemna minor* photosynthesis light reactions**: we measured photosynthesis of single *Lemna minor* fronds with a pulse amplitude modulation (PAM) fluorometer following standard protocols [15,16] using the saturating pulse mode over a time period of 30 min (Figure 2). As a reference, we used *Arabidopsis thaliana* plants grown under identical light intensity providing a comparison to a well-characterized terrestrial model organism of photosynthesis (Figure 2). We conducted measurements under actinic light intensities of 90 μmol photons m^−2^ s^−1^ (approx. growth light intensity) and around three-fold higher intensity (285 μmol photons m^−2^ s^−1^) to induce non-photochemical quenching processes. Dark-adapted *Lemna* fronds displayed a strong Kautsky effect after switching on illumination, which appeared highly comparable to equally treated and measured *Arabidopsis* plants. After the initial peak upon illumination, Chl fluorescence (F_t_) in *Lemna* declined rapidly and with the same velocity as in *Arabidopsis*, indicating that the onset of photochemical as well as non-photochemical quenching processes followed the same kinetics as far as was visible at the measured time scales. The Fv/Fm value (indicating the maximal quantum yield of PSII) was highly comparable between the plants under both actinic lights (close to 0.8, see Table 1). However, the steady-state Chl fluorescence F_s_ at the end of light acclimation in relation to F_m_ [17] was slightly lower in *Lemna* than in *Arabidopsis* (Table 1), suggesting a more efficient transfer of electrons from PSII into subsequent processes (either by photochemistry or dissipation). Interestingly, the NPQ value was clearly increased for *Lemna* under both actinic light intensities, suggesting a better excess energy quenching capability of the duckweed (Table 1).

These results suggested differences in photosynthesis efficiencies of *Lemna* and *Arabidopsis* and prompted us to compare photosynthesis parameters of both species in more detail. We performed a systematic analysis in which Chl fluorescence parameters of both plants (grown under identical light intensities) were determined at 12 different actinic light intensities ranging from 25 to 1500 μmol photons m^−2^ s^−1^ (Figure 3).

The F_v_/F_m_ values of *Lemna* and *Arabidopsis* were highly similar at low actinic light intensities (up to growth light intensity) but increasingly diverged in higher intensities. While *Lemna* exhibited rather stable values around 0.65–0.7, the values for *Arabidopsis* progressively declined, indicating that the photosynthesis apparatus of *Arabidopsis* had difficulties dealing with strong actinic light (Figure 3). The F_s_/F_m_ value for *Lemna* was found to be lower than that of *Arabidopsis* under all light intensities tested, indicating that the observed higher efficiency of excitation energy transfer (Figure 2) is a general property of *Lemna* rather than a punctual one appearing only under specific illumination conditions.

The higher efficiency in excitation energy transfer appeared to be caused by the combined action of photochemical and non-photochemical quenching processes because both the qP (the photochemical quenching coefficient) and qN (the non-photochemical coefficient) parameters were found to be higher in *Lemna* when compared to *Arabidopsis*. These results correspond well with the NPQ parameter that was strongly induced with light intensity in *Lemna* while it poorly developed in *Arabidopsis*. Besides these classical photosynthesis parameters, we also determined other parameters [18]. The Y.II parameter was found to be always higher in *Lemna* than in *Arabidopsis*, confirming the conclusion of improved excitation energy flux in *Lemna*. A striking difference between the two plants, however, was found for the Y.NPQ and Y.NO parameters. While the regulated energy dissipation (Y.NPQ) was largely identical for both, the unregulated energy dissipation (Y.NO) was much higher for *Arabidopsis* than for *Lemna*, suggesting that *Arabidopsis* loses more excitation energy as unregulated heat.

In sum, the duckweed *Lemna minor* exhibited a more flexible response to increasing actinic light intensities and was able to direct excitation energy more efficiently into photochemical work. At the same time, it presented a lower non-regulated dissipation of excess excitation energy. In all photosynthesis parameters tested in these standardized conditions, the duckweed clearly performed better than *Arabidopsis thaliana*.

**Analysis of thylakoid proteins and their phosphorylation state**: to obtain deeper insights into the photosynthesis apparatus of *Lemna minor* in comparison to *Arabidopsis thaliana*, we performed an initial structural characterization of photosynthesis thylakoid proteins by Western immunoblot analyses using commercially available antisera. To this end, we isolated total protein extracts from *Lemna* and *Arabidopsis*, separated them with SDS-PAGE and stained them with Coomassie (Figure 4A). The general protein pattern of the two species was of similar appearance, but we also observed many variations in size and accumulation of individual proteins which represent either different species-specific proteins or modifications of homologous proteins. The protein-specific Western immune analysis (Figure 4B) revealed that all photosynthesis proteins in *Lemna* can be readily detected with antisera directed against *Arabidopsis* proteins.

The general accumulation of these proteins was highly comparable between the two species, suggesting that the general stoichiometries of the protein complexes within the photosynthetic electron transport chain do not differ between *Lemna* and *Arabidopsis*. However, we observed a number of specific differences regarding the migration behavior of those proteins. While PSAA (the apoprotein A of PSI), ATPB (the beta subunit of the ATPase) and PSBA (the reaction center protein D1 of PSII) did not display visual migration differences, PSBC (the PSII core protein CP43), PSBO (the 33kD protein of the oxygen-evolving complex (OEC)), PETA (the cytochrome f subunit (Cytf) of the cytochrome *b_6_f* complex) and LHCB1 (one major chlorophyll A/B binding protein (CAB1) of the light-harvesting complex of PSII (LHCII)) all displayed a larger apparent molecular weight in *Lemna*. Only FNR (ferredoxin-NADP+-oxidoreductase) revealed a smaller one. In addition, the signal from *Lemna* PSBC appears to consist of two very closely running proteins. We checked available genomic information about the proteins displaying differential migration behavior to elucidate whether these differences could be caused by evolutionary sequence variations in the corresponding genes (Appendix A). We found, however, very high degrees of conservation between these *Lemna* and *Arabidopsis* photosynthesis proteins, indicating that the differences in migration behavior of the corresponding proteins cannot be simply attributed to sequence differences. This highly suggests the involvement of post-translational modifications of the respective proteins.

One major, if not the most important, post-translational modification of photosynthesis proteins is the phosphorylation at threonine residues in PSII proteins. Phosphorylation is further known to be a major cause for variation in the migration behavior of proteins separated by SDS-PAGE. In order to test whether variations in the phosphorylation status might cause the observed differences in the migration behavior, we examined the respective phosphorylation state of thylakoid membrane proteins by anti-phospho-threonine immune blotting. This technique is well established for terrestrial model plants but to our knowledge has never been employed in *Lemnaceae*. Our experiment was performed with material isolated from the light phase, a condition in which a high phosphorylation state of thylakoid membrane proteins should be expectable, especially in the growth light intensity we used. Indeed, we observed strong phosphorylation signals with both the *Arabidopsis* and the *Lemna* samples (Figure 5). However, we also identified clear differences in the phosphorylation patterns (indicated by arrows in Figure 5) that point to some specific differences in the phosphorylation states of both core as well as antenna proteins of PSII. These results support the idea that variations in post-translational modifications might be responsible for the observed differences in protein migration.

**Photosynthesis antenna movements and recovery from photoinhibition in *Lemna minor****:* phosphorylation of thylakoid membrane proteins is functionally connected to a number of physiological responses that adapt the photosynthetic apparatus to varying environmental illumination conditions. Phosphorylation of LHCB proteins in terrestrial plants is responsible for the dynamic and reversible association of LHCII to either PSII or PSI that optimizes light harvesting (a process called state transitions) [19], while the phosphorylation of PSII core proteins is known to be involved in the PSII repair cycle (also called recovery) after high-light-induced photoinhibition [20]. In order to assess the functionality of these processes in *Lemna minor*, we conducted specific physiological experiments that are known to trigger corresponding responses in *Arabidopsis* [21,22].

Phosphorylation of the LHCII in *Arabidopsis* is catalyzed by the redox-sensitive thylakoid kinase STN7 that becomes activated in light upon a reduced plastoquinone pool. Our light intensity conditions chosen for growth are ideal to activate this kinase. We thus performed 77K Chl fluorescence emission experiments with *Lemna minor* fronds (Figure 6) harvested from the dark phase (a condition with inactive kinase) and fronds harvested 50 min after the onset of white light (a condition expected to fully activate the kinase). In the dark sample, we observed a ratio for the Chl fluorescence emission at 735 nm (originating from PSI) to the emission at 686 nm (originating from PSII) of around 2.4. In the sample collected 50 min after onset of light, the ratio was raised to around 3.2, indicating that the relative antenna size of PSI largely increased upon white-light illumination. This light-induced increase in the F735/F686 ratio corresponds well to data reported for *Arabidopsis* [23], strongly suggesting that the *Lemna* homologue of *Arabidopsis* STN7 is working in a highly similar manner.

In order to assess the recovery capabilities of *Lemna minor* after sudden and severe light stress, we performed a high-light treatment of isolated fronds placed into wells of a micro-titer plate, each filled with liquid growth medium. To maximize the comparability of results, *Arabidopsis* plants in growth pots were placed alongside the *Lemna* fronds and at an identical distance to the light source (Figure 7). We used a high-intensity LED array that provided 1800 μmol photons m^−2^ s^−1^ white light at the height of the leaf surface to induce photo-damage of PSII. Light-induced photoinhibition typically becomes apparent by a decrease in the F_v_/F_m_ value since the D1 protein of PSII becomes damaged and less PSII reaction centers are available for photochemistry. The return of F_v_/F_m_ to normal levels then indicates the repair of the damaged PSII reaction centers. We recorded this Chl fluorescence parameter with a 2D imaging PAM in which both species can be assessed at the same time (Figure 7). Both *Lemna* and *Arabidopsis* displayed a decline of their F_v_/F_m_ values in response to a sudden and unprimed high-light stress treatment where the degree of decrease correlated with the length of the stress treatment (Figure 7B,C). Upon return to low-intensity growth light conditions, both plant species recovered to a full extent and with largely comparable kinetics (Figure 7C). In sum, our analyses indicate that the observed differences in the phosphorylation patterns of thylakoid membrane proteins of *Lemna* and *Arabidopsis* do not result in quantitative differences of phosphorylation-triggered acclimation responses of photosynthesis to varying illumination, at least under the conditions we have tested.

## 3. Discussion

The ecological niche of the aquatic *Lemnaceae* is highly different from those of terrestrial plants with respect to water and nutrient availability, light competition, herbivore attack and many more stressors. Despite these differences, the general photosynthetic properties of *Lemna minor* and *Arabidopsis thaliana* in terms of quenching kinetics and molecular organization appeared to be largely comparable in our study. We observed, however, some differences in the efficiency of excitation energy dissipation in our Chl fluorescence analyses that may have a long-term impact on ecological adaptation and species survival. In *Lemna gibba*, a close relative of *Lemna minor*, a high carotenoid content was identified as a major reason for the observation that this duckweed can easily deal with high light intensities [10]. If similar molecular responses also occur in *Lemna minor* will be elucidated in future studies. Our light intensity series experiment revealed that *Lemna minor* photosynthesis was superior to *Arabidopsis* photosynthesis also under weak actinic light, suggesting that further mechanisms (besides quenching by carotenoids) might contribute to the improved photosynthetic performance of *Lemna minor*. Thus, future studies including a wider range of physiological conditions will be required to allow for building a more sophisticated functional model of *Lemna minor* photosynthesis in comparison to terrestrial plants. The presented study, nevertheless, already provides a number of highly promising aspects on which such future studies can build on.

Our study reported here on *Lemna minor* provided a first analysis of selected properties known to be essential cornerstones of molecular photosynthesis regulation in terrestrial plants. Phosphorylation of threonine residues in thylakoid proteins by redox-controlled kinases is a well-characterized and described regulation mode that controls a number of photosynthesis responses to the environment [24]. Despite the differences in thylakoid membrane phosphorylation (Figure 5), *Lemna minor* displayed a well-conserved light-dependent association pattern of LHCII to the photosystems (Figure 6) and a kinetically comparable recovery response after photoinhibition with respect to the *Arabidopsis* control (Figure 7). The critical phosphorylation residues for these regulatory processes, thus, are likely conserved. The specific differences at the molecular level that were detected between the two species (Figure 4 and Figure 5) likely represent evolutionary adaptations to the different ecological niches of the two species. The detected differences in SDS-PAGE migration behavior are most likely caused by specific post-translational modifications because our bioinformatics analyses revealed that sequence variations of corresponding genes are very minor and do not cause size variations that could account for the observed migration differences.

The presented study was designed as a pilot project covering a number of basic photosynthesis parameters in order to elucidate whether more detailed analyses are worthwhile. The molecular differences we observed here are highly promising for our understanding of adaptations in plant photosynthesis to very diverse ecological niches and indicate a strong need for further and more detailed analyses. Such analyses should include, e.g., phospho-proteomics of thylakoid membrane proteins of *Lemna* isolated from plants subjected to varying and also extreme illumination and nutrient conditions. We hypothesize that the basic evolutionary protein chassis of C3 photosynthesis in plant species with highly differing ecological niches remains well conserved at the sequence level and that the required ecological adaptations to maintain photosynthesis efficiency as high as possible are achieved mostly by specific post-translational modifications. These are not restricted to phosphorylation alone and may also contain other modification types such as acetylation or glucosylation. More detailed studies of such modifications may provide important hints for improving indoor-farming approaches of terrestrial vegetables using aquaculture or for the plethora of biotechnology applications of *Lemnaceae*.

A side aspect of our study was the recording of high-magnification videos of *Lemna minor* growth in a sugar-free liquid medium. The major obstacle for such video recordings is the floating of individual fronds on the medium (eventually moving out of the camera focus) making it difficult to record the full time series of propagation in the long term. We solved this problem by using plastic grids as an anchor for the frond roots. While these grids largely immobilize the fronds, they still allow a free-floating life style and normal vegetative budding. A recent study reporting a comparable imaging approach used cell culture plates with agarose-solidified medium as growth source and anchor for *Lemna minor* fronds [25]. While this approach resulted in effective immobilization of fronds and successful recordings of *Lemna minor* multiplication, the resulting growth rates in sugar-free medium were relatively poor, suggesting that solid growth media are not beneficial for *Lemna minor* growth. Growth rates determined in our system were largely indistinguishable from reported growth rates in liquid media. Thus, our setup provides a useful tool to perform high-magnification live imaging of *Lemnaceae* growth that is close to natural conditions and opens up a technological possibility to test the influence of ions, molecules or inhibitors on *Lemna minor* growth in high resolution.

## 4. Materials and Methods

**Growth conditions and plant material.** *Lemna minor* (strain 9441) was obtained from the Appenroth collection at the Friedrich Schiller University of Jena, Germany. Duckweed colonies were grown under long-day conditions (8 h dark/16 h light) at 100 μmol photons m^−2^ s^−1^ white light provided by fluorescent stripe lamps (Lumilux Cool white L30W/840, Osram, Munich, Germany) at 21–24 °C. Plant colonies were pre-cultured and sterile in 1.5 L glass flasks sealed with cotton plugs (allowing for gas exchange) on nutrient media according to [13]. In order to maintain the cultures in the exponential growth phase, *Lemna* plants were regularly transferred to new flasks with fresh medium before they fully covered the available medium surface. For molecular and physiological analyses, plant material was either harvested by pouring the media with the *Lemna* colonies through a sieve (for collection of larger biomass) or single colonies were transferred by hand with an inoculation loop into specified vessels (for details see Results section) for in vivo Chl fluorescence measurements.

Wild-type *Arabidopsis thaliana* (var. Columbia-0) was grown on potting soil under light and temperature conditions identical to those of the *Lemna* culture. For molecular analyses, plant material of three- to four-week-old rosettes was harvested directly under the growth lights and was placed on ice or flash-frozen in liquid nitrogen and immediately used for further analyses. For in vivo Chl fluorescence measurements, plants were kept in their growth pots and placed under the emitter/detector unit (for details, see below) without any destructive manipulations.

**Growth rate determination by time-lapse video imaging.** *Lemna minor* fronds from the pre-culture were placed in a Petri dish equipped with a self-made, 3D-printer-generated grid and sealed with a gas-permeable tape under a digital microscope VHX 970F (Keyence, Osaka, Japan) (compare Figure 1C). Time-lapse videos were recorded using the camera’s internal timer program taking a picture every 15 min over the course of eight days, resulting in 758 photos. The total video duration is 25 s with a speed of 30 frames per second. During video recording, the fronds were maintained in a light/dark growth cycle comparable to the pre-culture by illuminating the Petri dish with two cold-white light sources (KL 1500 LCD, Zeiss, Oberkochen, Germany) with an approximate intensity of 100–150 µmol photons m^−2^ s^−1^. Photos were taken during a 10 s illumination by the camera’s internal white LEDs. For determination of the growth rate of *Lemna minor* fronds, the taken photos were digitally analyzed for the leaf-covered area using freely available ImageJ software.

**In vivo chlorophyll fluorescence measurements.** Room temperature chlorophyll fluorescence was detected by using a Junior pulse amplitude modulation (PAM) fluorometer (Walz, Effeltrich, Germany). For experimentation, individual fronds of *Lemna minor* were placed in a Petri dish containing growth media and immobilized by placing a weight on the tip of the roots to prevent plantlets from drifting on the liquid surface during the measurements. *Arabidopsis thaliana* (Col-0) plants grown on soil were measured for comparison. For both species, glass-fiber optics were placed 5 mm above the leaf surfaces in a way that the measuring light beam was centered on the leaf without touching the leaf edge. The intensity of the measuring light was adjusted to yield an approximate signal strength of 300 (rel. units) with non-dark-adapted plants. Plants were then dark-adapted for 30 min prior to each measurement. In initial experiments, actinic light intensity was set to either 90 or 285 µmol m^−2^ s^−1^ photons of PAR, and quenching analysis was performed using a custom-made script applying a saturation pulse every 20 s for a total duration of 30 min during the actinic light phase and increasing pulse width (20, 40, 80, 160 and 320 s) during the following dark recovery phase. Subsequently, a series of intensity measurements were conducted by increasing the actinic light intensity step-wise from 25 to 1500 µmol m^−2^ s^−1^ photons of PAR. A total of 12 different intensity steps were chosen by using the internal software control of the PAM device. Each step was preceded by a 30 min dark adaptation. All experiments were conducted in three independent biological replicates. Detected fluorescence values were calculated within the software of the PAM device to provide the yield of various photosynthetic parameters (see Results section). Standard deviations for all parameters were calculated and are given in Figure 3.

For recovery experiments, *Lemna minor* and *Arabidopsis thaliana* plants grown as described above were subjected to high-light (HL) illumination treatments using a custom-made 300 W LED Grow Light panel for the indicated time periods. The distance of the plants from the panel was adjusted to achieve a final PAR of approximately 1800 µmol m^−2^ s^−1^ photons at the leaf surface. For experimentation, individual fronds of *Lemna minor* were placed in 100 µL of growth media per well in black 96 multi-well plates (Greiner Bio-one, Kremsmünster, Austria) displaying a low auto-fluorescence and thereby minimizing interference with the Chl fluorescence signal detection.

In order to obtain the highest possible comparability between samples, individual leaves *of Arabidopsis* and individual wells with *Lemna fronds* were covered with light-tight aluminum foil before starting the high-light treatment. The foil was removed successively at defined time points from the plants until the total exposure time (indicated in the figures) was reached. Chl fluorescence of individual leaves and plants was detected in parallel using a 2D Imaging-PAM fluorometer (Walz, Germany). To ensure maximal consistency between experiments and organisms, the multi-well plate containing *Lemna* fronds and *Arabidopsis* plant leaves were adjusted to the same distance from the camera lens. Before exposure to HL stress, plants were first dark-adapted for 30 min, and F_v_/F_m_ was measured as a reference. After high-light treatment, plants were returned to their original growth light intensity and plants were allowed to recover from the light stress. F_v_/F_m_ was measured at three different time intervals (0 h, 2 h and 19 h) after HL treatment. To calculate the relative recovery, F_v_/F_m_ values obtained after HL treatment were expressed as a ratio to the reference values obtained before HL treatment. Each experiment was conducted with three independent biological and technical replicates (n = 3) and standard deviations were calculated.

**Western immunoblot analyses.** Freshly harvested material from *Lemna minor* and *Arabidopsis thaliana* was frozen in liquid nitrogen, homogenized in 1.5 mL Eppendorf tubes using pre-cooled pistils and mixed with an equal volume of 4 × SDS buffer (250 mM TRIS/ HCl pH 6.8; 40% (*w*/*v*) glycerol; 8% (*w*/*v*) SDS; 0.04% bromophenol blue; 5% (*v*/*v*) 2-mercaptoethanol) per weight ground plant material. The samples were incubated for 5 min at 95 °C. After centrifugation at 13,000 rpm for 10 min in an Eppendorf centrifuge, 5 µL of the supernatant containing the total soluble protein was separated by SDS–PAGE, using a 12% gel. For initial loading adjustment, the gel was stained with Coomassie and checked for sample comparability by judging the respective amount of lane loading. If required, protein loading was adjusted by volume variation. Adjusted protein amounts were then transferred to a nitrocellulose membrane (Amersham^TM^ Protran^TM^ 0.2 µm NC, 10600001) by semi-dry blotting. After blocking the membrane with TRIS-buffered saline (TBS), containing 5% *w*/*v* skim milk powder overnight at 4 °C, the membrane was washed twice with TBS-Tween (0.1%) and once with TBS for around 10 min each. The membranes were then incubated for 2 h at room temperature with primary antibodies and diluted in TBS, containing 5% milk powder (PSBA (Agrisera, Vännäs, Sweden, AS05 084) 1:10,000; PSAA (Agrisera, AS06 172), FNR (Agrisera, AS15 2909) 1:5000; PSBO (Agrisera, AS06 142-33) 1:5000; PSBC (Agrisera, AS11 1787) 1:3000; LHCB1 (Agrisera, AS01 004) 1:2000; ATPB (Agrisera, AS03 030) 1:5000; PETA (Agrisera, AS20 4377) 1:1000). Membranes were washed three times for 10 min in TBS-Tween (0.1%) and incubated for 1 h at room temperature with the secondary antibody and diluted in TBS, containing 5% milk powder (goat anti rabbit or a Lumi IgG coupled to horse radish peroxidase (Agrisera, AS09 602) 1:25,000; or rabbit anti-chicken IgY HRP (Promega Corporation, Madison, WI, USA, G1351) 1:10,000). After washing (three times for 10 min in TBS-Tween (0.1%) and twice for 10 min with TBS), the signal was detected using enhanced chemiluminescence (Lumi-Light^PLUS^ Western Blotting Substrate, Roche, Basel, Switzerland, or Pierce^®^ECL2 Western Blotting Substrate, Thermo Fisher Scientific, Waltham, MA, USA) and a BIORAD camera system or a Lumi Imager F1 (Roche). Signal detection times were adjusted in a way that sufficiently strong signals above background were achieved. *Lemna* and *Arabidopsis* samples typically showed comparable reactivity to the antisera except the one for the D1 protein. Here, the *Lemna* sample did react poorly, requiring an extension of exposition time to reach the same signal strength as the *Arabidopsis* sample. This, however, had no consequences for the data interpretation.

**Protein sequence alignments.** Alignments were performed using BLASTP (NCBI). Sequences of *Arabidopsis thaliana* proteins (TAIR, UniProt, Geneva, Switzerland) were used as a query. Standard parameters (standard databases, non-redundant protein sequences, threshold of 0.05, matrix BLOSUM62, cap costs existence: 11 and extension: 1 and conditional compositional score matrix adjustment) were used and it was blasted against *Lemna minor* (taxid:4472).

**The 77K fluorescence emission spectra.** The 77K low-temperature fluorescence measurements to obtain Chl fluorescence of PSI relative to PSII were conducted in a custom-made device using an LED source and a CCD detection array (Ocean Optics, Dunedin, FL, USA) corresponding to described setups [26]. During the whole measurement, the sample containing part of the device was kept in the liquid nitrogen. All wavelengths between 650 nm and 850 nm were measured. First, a blank was measured to ensure that the measuring cylinder was clean. Then, one leaf was harvested, clamped into the measuring device and directly incubated in liquid nitrogen to ensure that the leaf was completely frozen before the measurement was started. The measurement was repeated with the same leaf until two subsequent spectra resulted in the same values to ensure optimal measuring conditions. All recorded spectra were normalized to the average of the PSII Chl fluorescence emission peak at 686 nm. Three independent biological replicates for each condition were conducted and the standard deviation for the peaks at 686 and 735 nm were determined.

**Phosphorylation state of thylakoid proteins.** For the analysis of the phosphorylation state, thylakoid membranes were isolated according to [27]. Samples were homogenized in 40 mL ice-cold grinding buffer (50 mM HEPES/ KOH *pH* 7,5; 330 mM sorbitol; 2 mM EDTA; 1 mM MgCl_2_; 5 mM ascorbate; 0.05% (*w*/*v*) BSA; 10 mM NaF) using a waring blender. The homogenate was filtered through filter tissue (50 μm pore size), centrifuged at 5000× *g* and 4 °C for 4 min (Sorvall, Waltham, MA, USA, HB6) and the sediment was gently solved in 20 mL of shock buffer (850 mM HEPES/ KOH *pH* 7,5; 5 mM sorbitol; 5 mM MgCl_2_; 10 mM NaF). After incubation for 4 min, samples were centrifuged at 5000× *g* and 4 °C for 4 min (Sorvall, HB6) and the sediment was solved in 40 mL storage buffer (50 mM HEPES/ KOH *pH* 7.5; 100 mM sorbitol; 10 mM MgCl_2_; 10 mM NaF). Samples were further centrifuged at 5000× *g* and 4 °C for 4 min (Sorvall, HB6) and the sediment was solved in 750 µL or 1 mL storage buffer depending on the amount of fresh weight used. The chlorophyll concentration was measured according to [28]. Proteins were mixed with 4 × SDS sample buffer (0.075 M TRIS/ HCl *pH* 6.8; 6 M urea; 10% (*v*/*v*) SDS; 2% (*v*/*v*) glycerol; 0.5% (*v*/*v*) 2-mercaptoethanol; bromophenol blue) and denatured for 10 min at 68 °C. Amounts corresponding to 1 µg chlorophyll were separated by SDS PAGE (12%) and transferred to a nitrocellulose membrane (Whatman ^®^PROTAN Nitrocellulose Transfer Membrane 0.45 μm, GE Healthcare, Chicago, IL, USA). The membrane was blocked for 1 h at room temperature with 5% BSA dissolved in TBS-Tween (0.1%) and incubated with the primary antibody rabbit anti-phospho-threonine ((Cell Signaling Technology^®^, #9381) 1:1000), diluted in 2% (*w*/*v*) BSA in 1 × TBS-T (0.1%) overnight at 4 °C. The membrane was washed three times for 15 min in TBS-Tween (0.1%), incubated with the secondary antibody rabbit anti-chicken IgY HRP ((Promega Corporation, G1351) 1:10,000) for 2 h and again washed three times. For detection, ECL solutions I (0.1 M TRIS/ HCl pH 8.5; 2.5 mM Luminol (in DMSO); 0.04 mM p-coumaric acid) and II (0.1 M TRIS/ HCl pH 8.5; 5.4 mM H_2_O_2_) were mixed 1:2 and applied for 2 min to the membrane. Signal detection was conducted for 7 sec or 1 min, respectively. For verification of equal protein loading, the membrane was applied to amido black staining after ECL detection. The membrane was incubated for 5 min in the staining solution (50% (*v*/*v*) ethanol; 40% (*v*/*v*) water II; 10% (*v*/*v*) acetic acid; 4.06 mM Amido Black 10B) and de-stained in the de-staining solution (50% (*v*/*v*) ethanol; 40% (*v*/*v*) water II; 10% (*v*/*v*) acetic acid).

## Figures and Tables

**Figure 1 plants-12-02442-f001:**
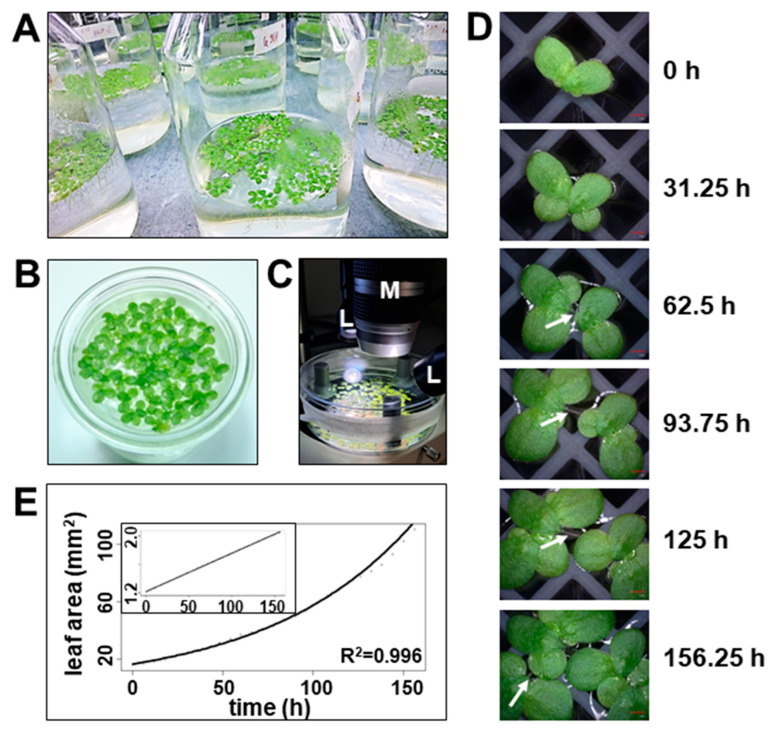
Imaging of *Lemna minor* growth characteristics in sugar-free liquid medium. (**A**) Sterile mass pre-culture of *Lemna minor* cultivated in flasks with standard growth medium. (**B**) *Lemna minor* fronds from exponential growth phase placed in vessel for detection of Chl fluorescence from top. (**C**) Set-up for time lapse movies recording *Lemna minor* growth. M: Objective of the digital microscope; L: Cold-white light source (**D**) Time series of growth of *Lemna minor* fronds taken by a digital microscope with automatic z-axis zoom and software-based image construction. Red scale bar in bottom right corner of images represents 1 mm. White arrows indicate the stalk connecting mother and daughter fronds. The complete time lapse video of this series is available in the supplement. (**E**) Growth curves of *Lemna minor* in sugar-free liquid medium determined as determined by the increase in leaf area detected in the time lapse video. The growth follows an exponential function (function of exponential regression: f(x) = 16.477e^0.0125x^). The inset in the upper left corner displays a semi-logarhythmic representation. ption.

**Figure 2 plants-12-02442-f002:**
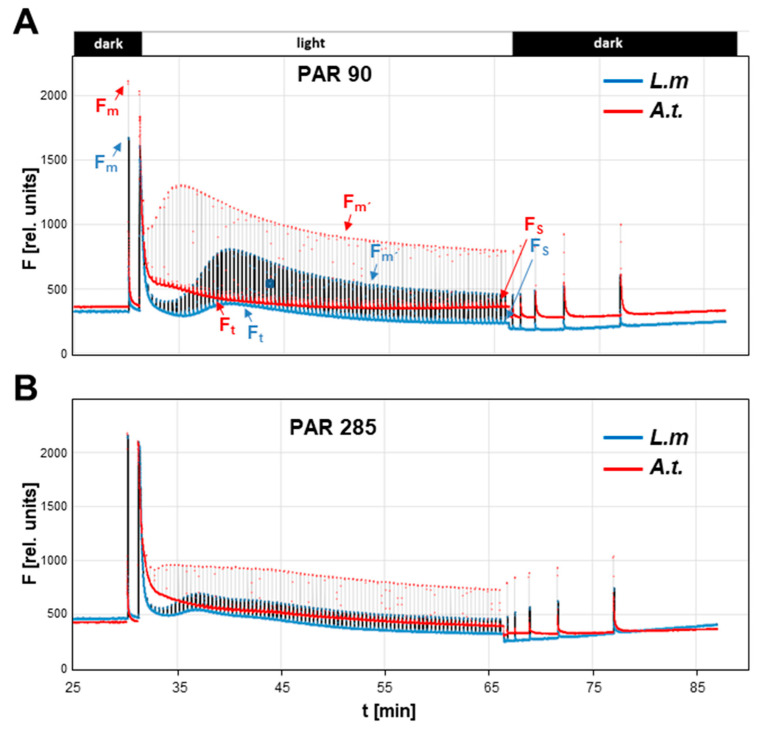
Photosynthesis characteristics of *Lemna minor* in comparison to *Arabidopsis thaliana.* Detection of Chl fluorescence in standard light quenching analyses of dark-adapted plants upon illumination using a Junior-PAM device operating in the saturation pulse mode. (**A**) Measurements in 90 µmol photons m^−2^ s^−1^ of actinic light. (**B**) Measurements in 285 µmol photons m^−2^ s^−1^ of actinic light. Fluorescence (F) is given in arbitrary units (rel. units). For direct comparison, fluorescence values of *Arabidopsis thaliana* at initiation of measurement were normalized to corresponding values of *Lemna minor*. Peaks in the records indicate maximal fluorescence upon application of saturation light pulses. Light and dark phases of actinic illumination are indicated on the top by white and black bars, the corresponding time scale is given on the bottom. The first 25 min of dark adaptation are not shown. One representative curve for each plant is given. For biological variations of parameters see Figure 3. Blue traces: Chl fluorescence of *Lemna minor (L.m.)*. Red traces: Chl fluorescence of *Arabidopsis thaliana (A.t.).* F_m_: Maximal fluorescence in the dark; F_m’_: Maximal fluorescence in the light; F_t_: Fluorescence in dependency of time t; F_s_: Steady state fluorescence. For details see text.

**Figure 3 plants-12-02442-f003:**
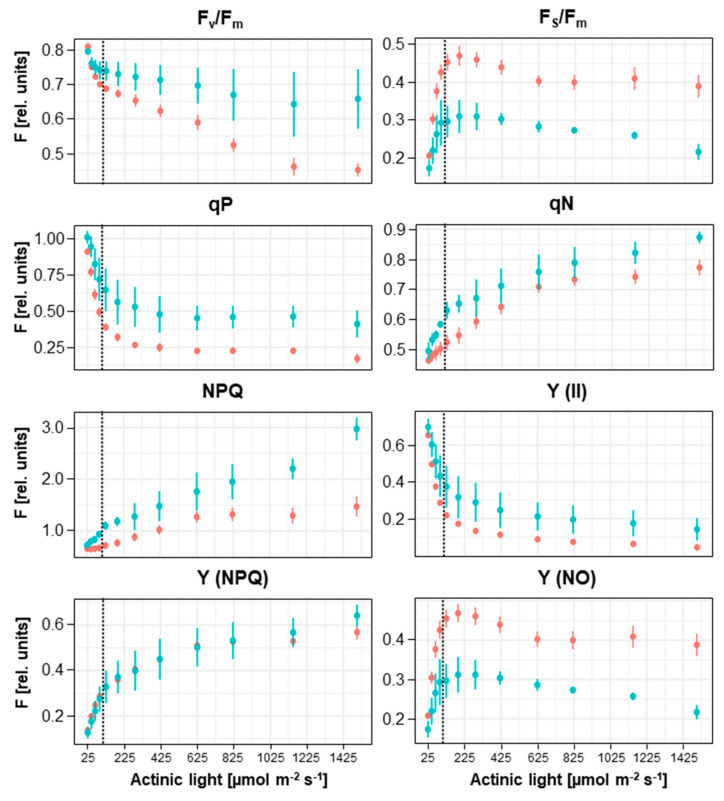
Comparison of photosynthesis parameters of *Lemna minor* and *Arabidopsis thaliana* under increasing actinic light intensities. Chl fluorescence of plants grown under identical light intensities was detected and analyzed with a Junior-PAM device as shown in Figure 2. Plants were measured consecutively in increasing actinic light intensities (given on the bottom). Dashed lines mark the growth light intensity. Identities of calculated photosynthesis parameters are indicated on the top of each graph, fluorescence is given in arbitrary units (left margin). Blue traces: Chl fluorescence of *Lemna minor*. Red traces: Chl fluorescence of Arabidopsis thaliana. For further details see text.

**Figure 4 plants-12-02442-f004:**
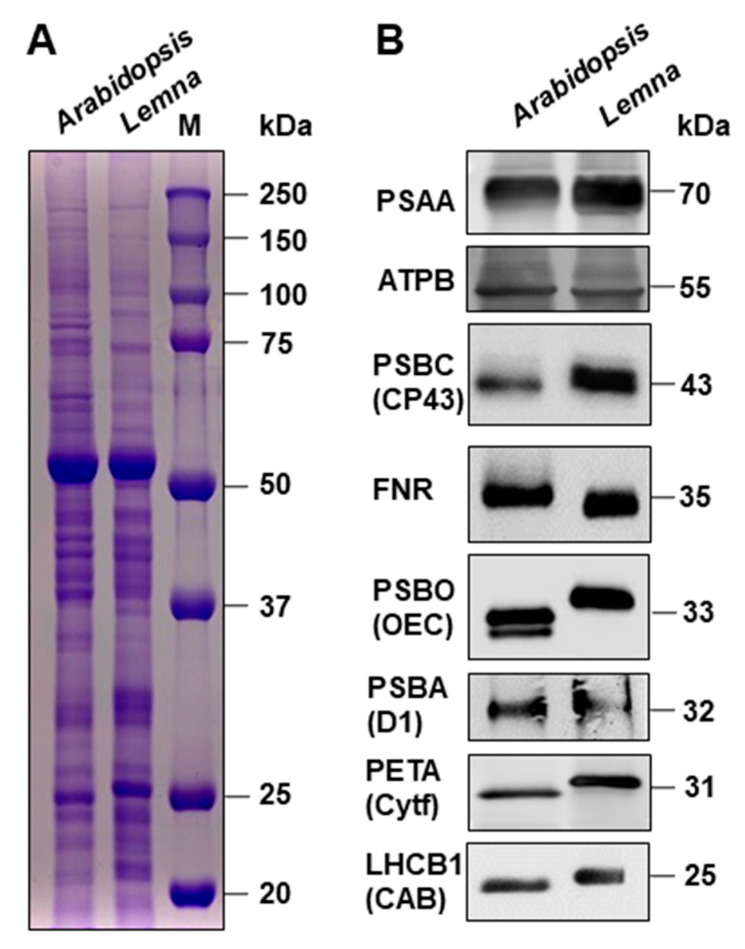
Comparison of total and photosynthesis proteins from *Arabidopsis thaliana* and *Lemna minor*. (**A**) SDS gel (12%) electrophoresis of total protein extracts isolated from white-light-grown plants (identities indicated on the top) stained with Coomassie. Marker (M) sizes are given in kDa in the right panel. (**B**) Western immunoblot analyses of selected photosynthesis proteins. Protein identities are indicated in the left margin using both gene-based and common (in parentheses) abbreviations, apparent molecular weight is given in the right margin. Loading of gel lanes was conducted as shown in A. Equality of loading was checked by Ponceau S staining of the membrane before immuno detection.

**Figure 5 plants-12-02442-f005:**
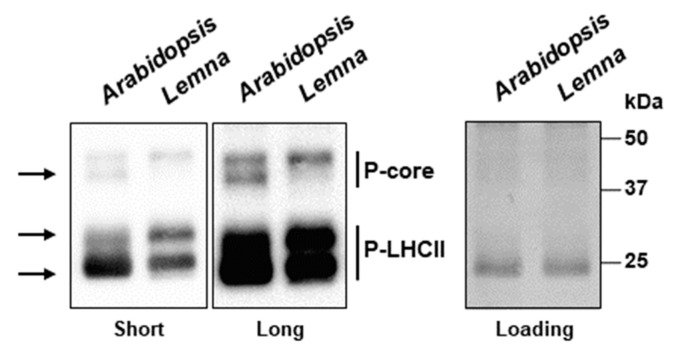
Phosphorylation state of thylakoid membrane proteins of *Lemna minor* in comparison to *Arabidopsis thaliana*. Protein amounts corresponding to 1 µg chlorophyll were separated by SDS PAGE and transferred to a nitrocellulose membrane via Western blot. The membranes were incubated with anti-phospho-threonine antibodies to detect phosphorylated thylakoid membrane proteins using enhanced chemiluminescence (ECL). Signal detection was conducted for 7 s (left panel, short) or 1 min (middle panel, long). Equal protein loading was tested by amido black staining of the membrane after ECL detection (loading, right panel). Sizes of marker proteins are given in the right margin in kDa. Phosphorylated LHCII and PSII core proteins are indicated in the right margin, differences in the phosphorylation signals arrows are indicated by arrows in the left margin.

**Figure 6 plants-12-02442-f006:**
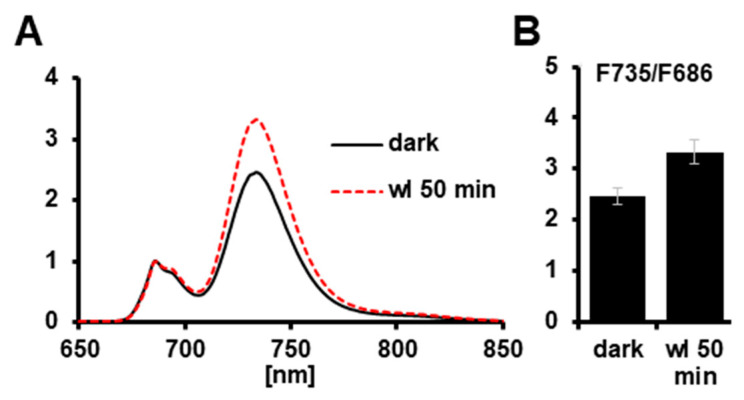
77K fluorescence spectra of *Lemna minor*. *Lemna minor* fronds were analyzed for Chl fluorescence emission in the range of 650–850 nm excitation in liquid nitrogen. Measured plant materials were harvested at the end of the dark phase of the growth light regime or 50 min after the onset of white-light illumination. All spectra were normalized to the fluorescence emission peak at 686 nm. (**A**) Representative spectra for both conditions. (**B**) Mean ratio of the fluorescence emission peaks at 735 and 686 nm (F735/F686) obtained from three independent replicates. SD is given.

**Figure 7 plants-12-02442-f007:**
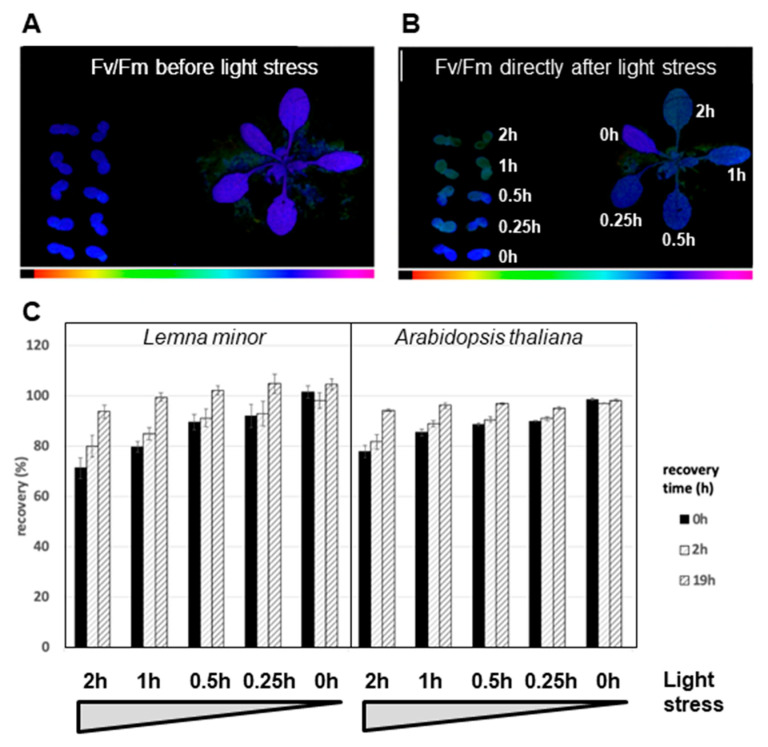
Recovery of photosynthesis in *Lemna minor* and *Arabidopsis thaliana* after high-light treatment. *Lemna minor (L.m.)* and *Arabidopsis thaliana* (*A.t.)* were exposed to high light (HL) (1800 µmol m^−2^ s^−1^ PPFD) and tested for recovery from photoinhibition. Note that this setup is different from the light intensity treatment given in Figure 3. (**A**,**B**) Representative images of the F_v_/F_m_ values recorded before (**A**, reference) and directly after (**B**) high-light (HL) treatment. Duration of illumination for the HL treatment is indicated by the numbers in **B**. F_v_/F_m_ values are color-coded from black indicating F_v_/F_m_ = 0 to purple indicating F_v_/F_m_ = 1 (see color gradient bar below the images). (**C**) Duration of HL stress is indicated on the x-axis beginning with the strongest stress. F_v_/F_m_ was measured before the HL treatment (as reference) and immediately (0 h, black bars), 2 h (light grey bars) and 19 h (dark grey bars) after return to growth light intensity. Recovery (rec. %) is expressed relative to the reference F_v_/F_m_ measured on an individual leaf/frond base. Results are averages of three biological independent replicates. Small black vertical bars indicate the standard error of the mean (SEM).

**Table 1 plants-12-02442-t001:** Comparison of selected Chl fluorescence parameters of *Lemna minor* and *Arabidopsis thaliana* calculated from the measurements given in Figure 2.

Actinic Light	Species	Fv/Fm *	Fs/Fm *	NPQ *
90 μmol m^−2^ s^−1^	*L. minor*	0.809	0.14	2.6
	*A. thaliana*	0.799	0.20	1.5
285 μmol m^−2^ s^−1^	*L. minor*	0.783	0.15	3.7
	*A. thaliana*	0.796	0.19	1.9

* Fv/Fm: Variable fluorescence in relation to maximal fluorescence in the dark representing the maximal quantum yield of PSII; * Fs/Fm: Steady-state fluorescence in the light in relation to maximal fluorescence in the dark representing a measure for the energy transfer efficiency from light reactions to subsequent processes; * NPQ: Quenching of absorbed light energy by non-photochemical processes. Values represent single calculated data from original traces shown in Figure 2. For variations between independent biological repetitions, see Figure 3.

## Data Availability

The data presented in this study are available in the article itself and the Appendix A accompanying the article.

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
