# Peer review of "Photosynthesis in the Biomass Model Species Lemna minor Displays Plant-Conserved and Species-Specific Features"

_plants, 2023, doi:10.3390/plants12132442_

Round 1

Reviewer 1 Report

This is an interesting article that reads well. 

It might be relevant to include a paragraph on duckweed evolution from terrestrial plant to aquatic plant.

There are some minor problems with English. For instance: top of page 2 – “sweet water ponds” ?? Is there sugar in the ponds or do authors mean “freshwater”?

Author Response

Response to reviewers

Reviewer 1

This is an interesting article that reads well. It might be relevant to include a paragraph on duckweed evolution from terrestrial plant to aquatic plant.

Response: We added a few comments about the evolution of the reduced morphology of duckweeds in the introduction.

There are some minor problems with English. For instance: top of page 2 – “sweet water ponds” ?? Is there sugar in the ponds or do authors mean “freshwater”?

Response: We apologize for this „germanized“ term; yes, of course „freshwater“ is meant. That is corrected. Further, we went carefully through the manuscript and checked spelling and wording.

Reviewer 2 Report

The manuscript is focused on the description of photosynthesis–related features specific to the species Lemna minor, a representative of aquatic angiosperms. Lemna is compared to a model terrestrial angiosperm Arabidopsis thaliana grown at the same light conditions. Except of the widely known fast growth of Lemna, the authors have found the following differences compared to Arabidopsis:  i) Lemna has slightly different photosynthetic induction (measured by modulation chlorophyll fluorimetry), ii) some photosynthesis-related proteins (Lhcb1, PetA, PsbO) have posttranslational modification based on the differences in the migration distance of proteins identified by immunoblotting and ii) Lemna has more pronounced phosphorylation of LHCIIs and PSII core complexes based on phosho-imunoblotting. All other presented characteristics of Lemna were similar to those of Arabidopsis. The authors present these results as a pilot study describing the “most likely evolutionary adaptations to different ecological niches” of Lemna. Unfortunately, I have to state that this conclusion is very speculative as it is based on fragmented data without interpretation.

The data to i) consist of only one measurement for Arabidopsis and Lemna leaves for one actinic light intensity. There is no statistics, no evaluation of quenching parameters. Moreover, the induction took just 6 min, i.e. the evaluated Fs level may not represent the steady state fluorescence level.  Further, I cannot understand how the authors calculated the values Fs/Fm (see the sentence in the text: “However, the steady state Chl fluorescence parameter Fs [17] in relation to Fm was slightly lower in Lemna than in Arabidopsis (0.06 in Lemna versus 0.21 in Arabidopsis)”). The values do not correspond with Figure 2. In addition, one have to take into account that the fluorescence induction is a very complex signal and could reflect e.g. actual gas exchange determined by the extent of stomata openness. A question arises what are the gas exchange characteristics of Lemna, which has the stomata only on the upper leaf surface. Much more work has to be done to get useful information from the fluorescence induction measurements.

The data to ii) are purely descriptive. There is no information on what kind of modification appears in Lemna and in particular, what should be the relevance of the data to the “most likely evolutionary adaptations to different ecological niches” stated by the authors.

The data to iii) should be completed by the evidence that the phosphorylated proteins are really PSII cores and LHCIIs. The migration distance is not enough for this attribution. Again, there is no relevance of the data to the stated adaptation of Lemna to the growth on water.

In conclusion, the manuscript describes some differences in Lemna and Arabidopsis in some photosynthesis-related characteristics, however, it is not clear whether they are the ones that are crucial for the adaptation to land or water niches. The presented data are very preliminary and should serve as basis for more comprehensive work, which could indeed improve our knowledge of important factors determining the differences in the physiology of Lemna and Arabidopsis.

Author Response

Response to reviewers

Reviewer 2

The manuscript is focused on the description of photosynthesis–related features specific to the species Lemna minor, a representative of aquatic angiosperms. Lemna is compared to a model terrestrial angiosperm Arabidopsis thaliana grown at the same light conditions. Except of the widely known fast growth of Lemna, the authors have found the following differences compared to Arabidopsis:  i) Lemna has slightly different photosynthetic induction (measured by modulation chlorophyll fluorimetry), ii) some photosynthesis-related proteins (Lhcb1, PetA, PsbO) have posttranslational modification based on the differences in the migration distance of proteins identified by immunoblotting and ii) Lemna has more pronounced phosphorylation of LHCIIs and PSII core complexes based on phosho-imunoblotting. All other presented characteristics of Lemna were similar to those of Arabidopsis. The authors present these results as a pilot study describing the “most likely evolutionary adaptations to different ecological niches” of Lemna. Unfortunately, I have to state that this conclusion is very speculative as it is based on fragmented data without interpretation.

The data to i) consist of only one measurement for Arabidopsis and Lemna leaves for one actinic light intensity. There is no statistics, no evaluation of quenching parameters.

Response: The original focus of the study was on the molecular data, the physiological data were added to provide just a glimpse on the vitality of the two species under the used growth conditions and to demonstrate that both are healthy and perform photosynthesis without any problems.

We agree with the reviewer that a more comprehensive and sophisticated Chl fluorescence study provides a much better base of data for physiological conclusion. We thus improved Fig. 2. We did new measurements of Arabidopsis and Lemna (3 biological replicates each) and plotted representative Chl fluorescence data (actually we did not observe any significant differences between the individual measurements) into the same graph for direct comparison of the fluorescence traces and the resulting steady state fluorescence. We also provide a new table (Table 1) with important fluorescence parameters and their variations.

Further we performed additional experiments and provide now a detailed Chl fluorescence study (new Figure 3) for Lemna minor and Arabidopsis (both grown under identical light intensities). We determined various Chl fluorescence and quenching parameters under 12 different actinic light intensities (ranging from 25 – 1500 µE) done in 3 independent biological replicates allowing to perform statistical evaluation of data. With this new data we can demonstrate that Lemna indeed is able to respond much more flexible to increasing actinic light intensities than Arabidopsis, mostly by directing more energy into photochemical work, but also by dissipating excess excitation energy more effectively. This detailed analysis uncovered that especially the yield NO (the unregulated dissipation by heat) is much smaller in Lemna demonstrating that the duckweed is losing less of the absorbed energy. In fact, the duckweed performed superior to Arabidopsis in all parameters determined.

Moreover, the induction took just 6 min, i.e. the evaluated Fs level may not represent the steady state fluorescence level.  Further, I cannot understand how the authors calculated the values Fs/Fm (see the sentence in the text: “However, the steady state Chl fluorescence parameter Fs [17] in relation to Fm was slightly lower in Lemna than in Arabidopsis (0.06 in Lemna versus 0.21 in Arabidopsis)”). The values do not correspond with Figure 2.

Response: We have done now new measurements that lasts at least 30 min and the Fs/Fm values have been re-calculated from these to provide a consistent comparison with the other Chl fluorescence parameters. These are included in the new Table 1 as well as their statistical variation. In addition, Fig. 2 has been re-drawn. The calculation of Fs/Fm is given in detail in reference 17.

In addition, one have to take into account that the fluorescence induction is a very complex signal and could reflect e.g. actual gas exchange determined by the extent of stomata openness. A question arises what are the gas exchange characteristics of Lemna, which has the stomata only on the upper leaf surface. Much more work has to be done to get useful information from the fluorescence induction measurements.

Response: Yes, we fully agree, fluorescence data are influenced by a plethora of parameters. Gas exchange measurements for sure are interesting and we currently try to establish a measurement set-up, which, however, is technically very challenging since the duckweeds need to swim on medium that buffers the amount of released CO2. Thus, we need to shift these experiments to future studies. With the much more extended Chl fluorescence study we, however, can provide a clearly improved interpretation of the photosynthetic processes in Lemna also without gas exchange measurements.

The data to ii) are purely descriptive. There is no information on what kind of modification appears in Lemna and in particular, what should be the relevance of the data to the “most likely evolutionary adaptations to different ecological niches” stated by the authors.

Response: Yes, indeed, these data are descriptive, that is the central informational value of western analyses. They describe (for the first time to our knowledge) a direct comparison of Arabidopsis and Lemna samples separated on the same gel probed with the same antisera. Of course this is only a starting point for further studies. With these experiments we aimed to test whether doing a sophisticated in-depth proteomics study is justified. This we now can definitely confirm and include a sentence in the discussion that proteomic studies will be done in future. Concerning the “evolutionary adaptations”. This is our interpretation why we observe the migration differences. We regard this indeed as likely since the primary amino acid sequences are largely identical as far as we could see. Since it is of course of speculative nature we toned down this interpretation and indicate it as one possible way, without excluding other alternatives.

The data to iii) should be completed by the evidence that the phosphorylated proteins are really PSII cores and LHCIIs. The migration distance is not enough for this attribution. Again, there is no relevance of the data to the stated adaptation of Lemna to the growth on water.

Response: In this assay we have separated isolated thylakoid membrane protein fractions (not total proteins). Further, the used antiserum is phosphothreonine-specific and is well-known to detect only LHCII and PSII core proteins in the thylakoids since only here phosphothreonine residues are found. Actually, it is a highly accepted standard technology for that kind of study. Our sequence analyses have further revealed that the Lemna PS protein sequences are highly conserved in comparison to Arabidopsis, it is thus conceivable to conclude that the signals belong to these proteins. We, however, understand the concern of the reviewer that there is the theoretical possibility that phosphorylated threonine residues in completely other proteins (this would be novel threonine phosphorylation sites) could be detected. We thus toned down our conclusion and discuss it as “likely”.

In conclusion, the manuscript describes some differences in Lemna and Arabidopsis in some photosynthesis-related characteristics, however, it is not clear whether they are the ones that are crucial for the adaptation to land or water niches. The presented data are very preliminary and should serve as basis for more comprehensive work, which could indeed improve our knowledge of important factors determining the differences in the physiology of Lemna and Arabidopsis.

Response: We now provide much more comprehensive data that unequivocally demonstrate that Lemna displays much more flexible and effective photosynthesis parameters. Whether these improvements are cause or consequence of the aquatic life style is difficult to answer, however, duckweeds have very well adapted to their ecological niche and a corresponding adaptation of photosynthesis is conceivable, we discuss this aspect now more carefully. Also the abstract was improved correspondingly.

Reviewer 3 Report

The manuscript provides valuable information for explaining the high biomass accumulation in duckweed, and a useful approach for the studying of the growth-related parameters in duckweed. And It is helpful to investigate the photosynthesis mechanism in duckweed.

However, there are some problems in the manuscript, please revise them.

1. About languages, some of them have been marked in the manuscript in yellow. Please check the whole manuscript.

2.About Figures, there are no significance analysis in Figure 5 B and Figure 6 C. And some figures are not clear, for instance Figure 6. And the Lemna and Arabidopsis should write in italic in Figure 6 C.

3. The writing style is not same in the reference, for example, some of the first letter of the word are in capital, some are in lowercase letters in the titles; Some species didn’t write in italic.

Author Response

Reviewer 3

The manuscript provides valuable information for explaining the high biomass accumulation in duckweed, and a useful approach for the studying of the growth-related parameters in duckweed. And It is helpful to investigate the photosynthesis mechanism in duckweed.

However, there are some problems in the manuscript, please revise them.

  1. About languages, some of them have been marked in the manuscript in yellow. Please check the whole manuscript.

Response: We did the requested improvements and went through the manuscript for further improvements. We also reworded the abstract and various parts of the manuscript (highlighted in yellow in the revised version).

2.About Figures, there are no significance analysis in Figure 5 B and Figure 6 C. And some figures are not clear, for instance Figure 6. And the Lemna and Arabidopsis should write in italic in Figure 6 C.

Response: In Fig. 5B and Fig. 6C the standard deviation is given as bar, resolution of Fig. 6 was improved and the species names were set in italics. Fig. 6C is maybe a bit tricky because there are two time dependencies, that of the initial photoinhibition and that of the recovery. We show the recovery from the strongest photoinhibition first, that causes a reverse arrangement of times. We improved the labelling to make this clear.

  1. The writing style is not same in the reference, for example, some of the first letter of the word are in capital, some are in lowercase letters in the titles; Some species didn’t write in italic.

Response: This is caused by the reference program that takes the references from the sources as they are. We improved this manually.

Round 2

Reviewer 2 Report

The manuscript has been suffciently improved according to my remarks.